

# On the correlation between ROTI and S4

Yang Liu[1], Sandro Radicella[2]

[1] School of Instrumentation Science and Opto-Engineering; Beihang University, Beijing, 100191, P.R.China
[2] Abdus Salam International Centre for Theoretical Physics, Telecommunications/ICT for Development Laboratory, Trieste,
34151, Italy

*Correspondence to*: Yang Liu (liuyangee@buaa.edu.cn)

**Abstract.** The correlation between Rate of TEC index (ROTI) and amplitude scintillation index S4 has been under focus for years, since both of the two are associated with ionospheric irregularities. Previous studies show that the behavior of the correlation between ROTI and S4 is not regular. To address this problem, in the present work the relation between S4, ROTI and the Differential Rate of TEC index (DROTI) has been investigated, assuming the single screening model for trans-ionospheric radio propagation. The influences of the effective velocity and elevation angle and their variability on the correlation between ROTI and S4 are analyzed. Data from two stations of Scintillation Network Decision Aid (SCINDA) network are used for the study. Results show that the variability of effective velocity plays a crucial role on the correlation between the amplitude scintillation index and the TEC rate of change indices. Based on the Gaussian assumption of effective velocity distribution, a high correlation coefficient can be achieved only under low variability conditions.

## 1 Introduction

Ionospheric irregularities usually refer to small or medium scale structures in the ionosphere plasma and the occurrence of plasma bubbles. Trans-ionospheric signals passing through them suffer from scintillation, described by signal amplitude and phase alteration (Yeh and Liu, 1982). One of the most used ways to measure ionosphere scintillation is its effect on GNSS signals. Scintillation intensity can be observed by calculating the GNSS signal variability, the more severe scintillation is, the larger scintillation indices will be (Cervera et al, 2006; Kintner et al, 2007). Ionospheric irregularities can be assessed through the rate of change of total electron content (TEC) indicated by ROT, which can be derived also from dual frequency GNSS receiver. Pi et al (1997) proposed the ROT index ROTI as indicator of ionospheric irregularities. Being ionospheric scintillation of radio signals an important phenomenon under focus, the relationship between ROTI and scintillation indices has been widely studied. Basu et al (1999) noted that ROTI/S4 varies in correspondence to the zonal velocity of ionospheric irregularities. Yang and Liu (2016) considered each satellite-receiver propagation path, and found larger correlation between ROTI and S4 can be achieved under stronger scintillation circumstances. The study also found that more often poor correlations between them exists. Liu et al (2019) further addressed the impacts of receivers and sampling rates on the behaviours of ROTI. Carrano et al (2019) showed evidences that only under certain geometry ROTI and S4 are highly correlated., the same study clarifies the differences between ROTI and S4 considering that S4 is more influenced by the



scintillation spectrum and Fresnel scale (Li et al ,2007). A signal propagation model proposed by Du et al (2000) tried to relate ROTI with S4 with fixed effective velocity; the model was later again used by Amabayo et al (2015) to investigated the correlation between derived S4 from Du's model and the observed S4; their results show stronger correlations under some geographical locations. The limitation of Du's model as pointed out by Carrano et al (2019) is mainly the lack of

consideration about detailed scintillation spectrum parameters; while it has been found that effective velocity is related to both the velocity of ionospheric pierce point (IPP) and zonal velocity (Arruda et al, 2006; Chapagain et al, 2013). The assumption of constant effective velocity influences the correlation between ROTI and S4..

To further investigate this problem, the present work starts from the single layer phase screen model to derive the relationship between ROTI and S4; a formula of S4 is proposed in relation with the differential rate of TEC index (DROTI),

the effective velocity and geometry. The correlation between ROTI and S4 are further studied, considering the randomity of effective velocity. Section 2 gives the basic derivation of ROTI and the proposed methodology; Section 3 introduces the data used for experiments; Section 4 and 5 show the results and discussions; Finally, some conclusion are given.

## 2 Methodology

### 2.1 ROTI calculation

TEC is derived by extracting ground GNSS observables from SCINDA files. For each slant path between satellite and receiver on the ground, a slant TEC can be calculated assuming that the ionosphere is concentrated in the thin shell at a given height. To describe the disturbance of ionosphere TEC, rate of change of slant TEC is calculated. In addition, an index of the rate of slant TEC, which considers the average standard deviation of slant TEC rate has been proposed, and referred as ROTI by Pi et al (1997). Considering that the slant TEC is given by

$$ROT = \frac{STEC_{k+1} - STEC_k}{\Delta t_k},\qquad(1)$$

where $STEC_{k+1}$ and $STEC_k$ are slant TEC at $k+1$ and $k$ time epochs; $\Delta t_k$ is time interval; usually the unit of ROT is TECU/min. ROTI has been defined by Pi et al (1997) as:

$$ROTI = \sqrt{\langle ROT^2 \rangle - \langle ROT \rangle^2},\qquad(2)$$

where $\langle ROT \rangle$ denotes averaging ROT during $N$ epochs. Experimentally, a threshold of 0.2 TECU/min is set to determine

whether irregularity exists (Pi et al, 1997). The relation between ROTI and scintillation indices has already been studied to verify if ROTI as a reliable indicator of ionosphere irregularity (Bhattacharyya et al, 2001; Yang and Liu, 2016). The index DROTI takes the standard deviation of DROT instead of $ROT$, which is the second order TEC variation, given as

$$\frac{dROT}{dt} = DROT,\qquad(3)$$

Substituting $DROT$ with $ROT$ in equation (2), we get the expression of DROTI

$$DROTI = \sqrt{\langle DROT^2 \rangle - \langle DROT \rangle^2},\qquad(4)$$



## 2.2 Proposed method

In one-dimensional phase screen model, assuming that phase varies only along the x-direction, and according to the transport of intensity equation (Rino 1979), the following equation can be obtained:

$$\frac{d^2\phi}{dr^2} = -\frac{2\pi}{\lambda s}\left[1 - \frac{I(r,s)}{I_0}\right], \tag{5}$$

where $I$ is the unified signal amplitude intensity, $I_0$ is its standard level under non-scintillation conditions. $\lambda$ denotes the signal wavelength. Further, the phase variation $\phi(x)$ is related to $TEC(x)$ along the signal path, according to the phase screening theory:

$$\phi(x) = -\lambda r_e TEC(x), \tag{6}$$

$r_e = 2.8 \times 10^{-15}$m is the radius of the electron. Spatial variations along x-direction are converted to temporal variations at

the site of the receiver due to the relative motion of the irregularities with respect to the signal path. Then (5) was rewritten as:

$$\frac{d^2 TEC(r)}{dt^2} = \frac{2\pi}{\lambda r_e}\frac{v^2}{\lambda s}\left[1 - \frac{I(r,s)}{I_0}\right], \tag{7}$$

The left side of (3) approximates to the variation of ROTI, represented as DROTI, and the right side approximates to the amplitude scintillation S4 (see **Appendix**). Therefore, from the above equation it can be written:

$$v^2 = DROTI \cdot \frac{\lambda^2 r_e s}{2\pi S_4}, \tag{8}$$

where $s$ is the slant distance between phase screen ionosphere pierce point and the GNSS receiver. $R_e$ denotes the radius of the earth, $h_{IPP}$ denotes the height of ionosphere pierce point from earth, usually considered 450km. $v$ is the effective velocity described in single phase screen model (Carrano et al, 2016; Rino, 1979). The velocity is related to ROTI variation, amplitude scintillation index S4 and scintillation intensity. To simplify the equation, it gives that

$$S_4 = DROTI \cdot \frac{\lambda^2 r_e s}{2\pi v^2}, \tag{9}$$

## 3 Data

In this work, the GPS RINEX and ionospheric scintillation monitor (ISM) observables were used. The observables were provided by two stations from the Air Force Research Laboratory Scintillation Network Decision Aid (AFRL-SCINDA) network, dedicated to monitor low latitudes ionospheric scintillations. The observables provided directly TEC and

scintillation indices (amplitude scintillation index S4 and phase scintillation index $\sigma_\phi$). September 2013 (a relatively quiet geomagnetic period) data from two SCINDA stations: Cape Verde (CVD, 16.7°N, 23.0°W) and Dakar (DKR, 14.7°N, 17.5°W) were chosen. The observables were collected from NovAtel GSV4004B scintillation monitors. The amplitude scintillation index S4, that takes the standard deviation of amplitude scintillation intensity is used. To avoid the interference of geomagnetic perturbations, the Dst indices of September 2013 are shown in Figure 1.



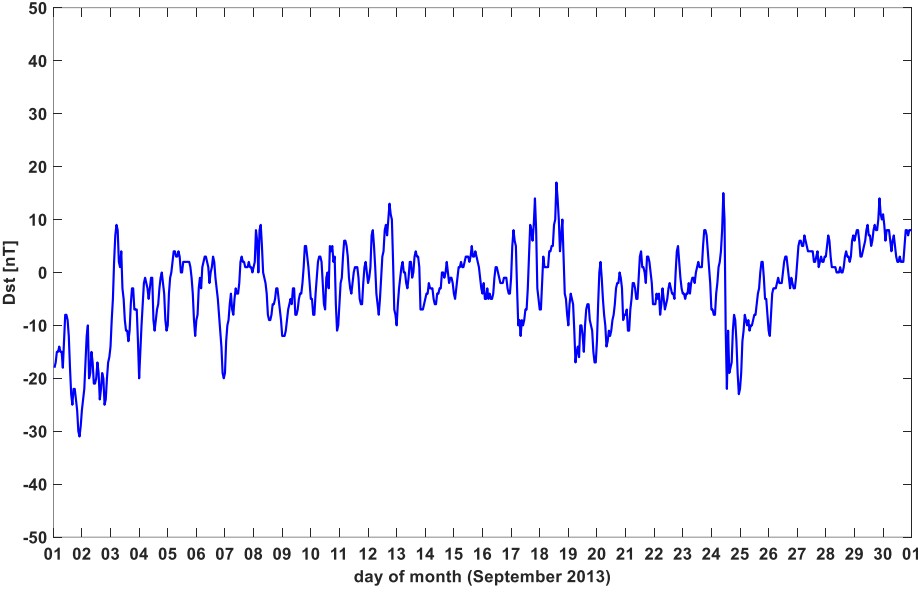

**Figure 1: Dst index in September 2013.** the values are above -50 nT, indicating no geomagnetic storms happened in the month. The sampling rate of TEC and scintillation indices is 1 minute, to further calculate ROTI index proposed by Pi et al. (1997), a five-minutes window span was used with 30° elevation mask to exclude multipath interference. Several scintillation events from CVD station were selected and shown in Figure 2. Three cases are shown for PRN 15 satellite.





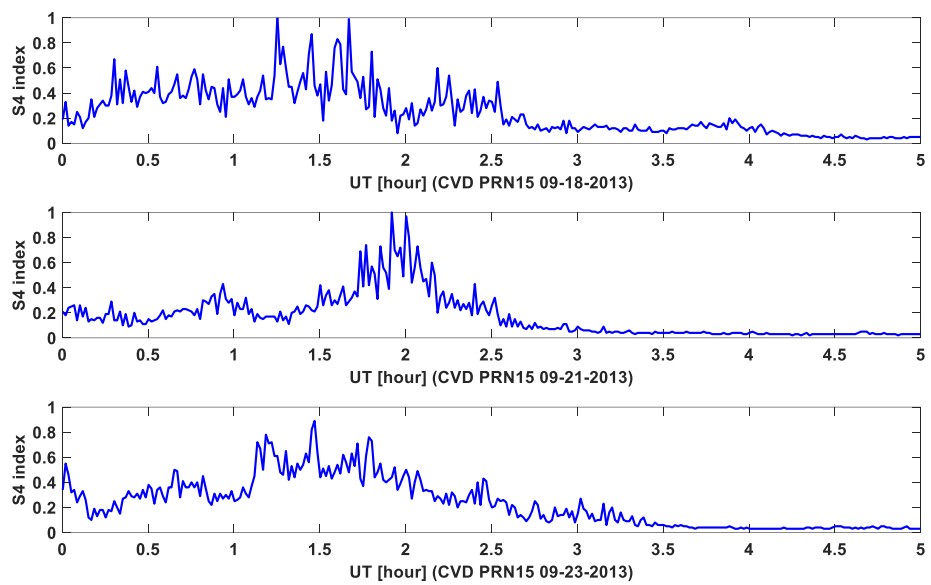

**Figure 2: Scintillation events selected from CVD station.** Three cases were shown for PRN 15 satellite. Strong amplitude scintillations were noticed, from 00:00 UT to 03:00 UT with S4 approaching to 1. The effective velocity is calculated with equation (8) for both CVD and DKR stations It shows that the effective velocity has daily variabilities.

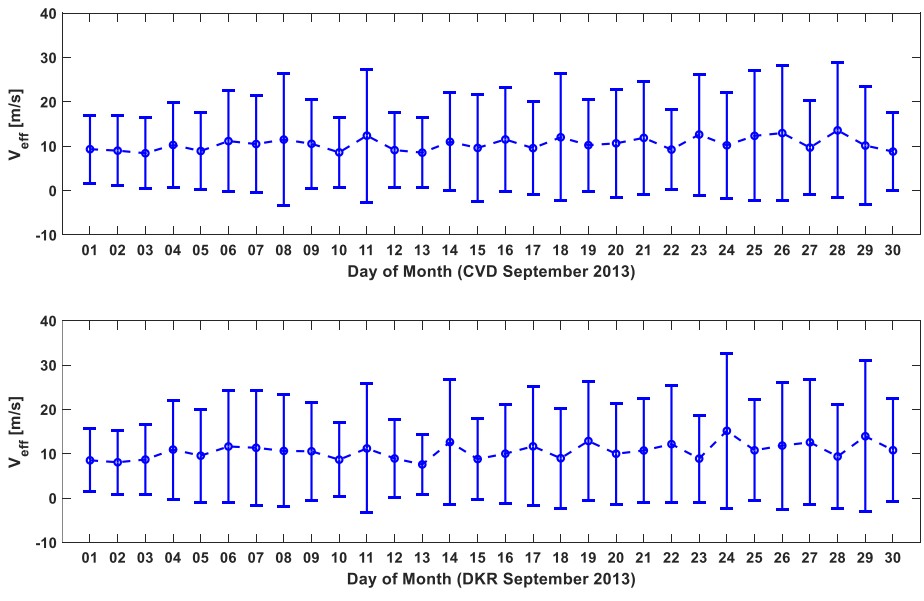




**Figure 3: Daily variability of effective velocity for both CVD and DKR stations.** The effective velocity has mean value around 10 m/s, but with large standard deviation, varying from 10 m/s to 30m/s.

## 4 Experimental results

### 4.1 Correlation between ROTI and observed S4

The correlation between ROTI and observed S4, which was calculated from the observables is shown in Figure 4. To better feature the correlation, a normalized ROTI was used by dividing all ROTI values with the maximum ROTI in the month (Yang and Liu. 2016). The correlation between ROTI and S4 looks very poor in this case, despite some stronger scintillation event existed in the same month. Most S4 index values are lower than 0.5, only ~20 cases indicate severe scintillation events, with S4 above 0.7. It should be noted that in some cases large S4 corresponded to small normalized ROTI; similarly, in some

cases small S4 corresponded to large normalized ROTI. The overall correlation coefficient was 0.555 for Cape Verde and 0.566 for Dakar indicating a low correlation between ROTI and S4. The results indicate that large normalized ROTI with small S4 and similarly large S4 with very small ROTI can be found, showing that ionospheric scintillations are not directly related to ROTI.

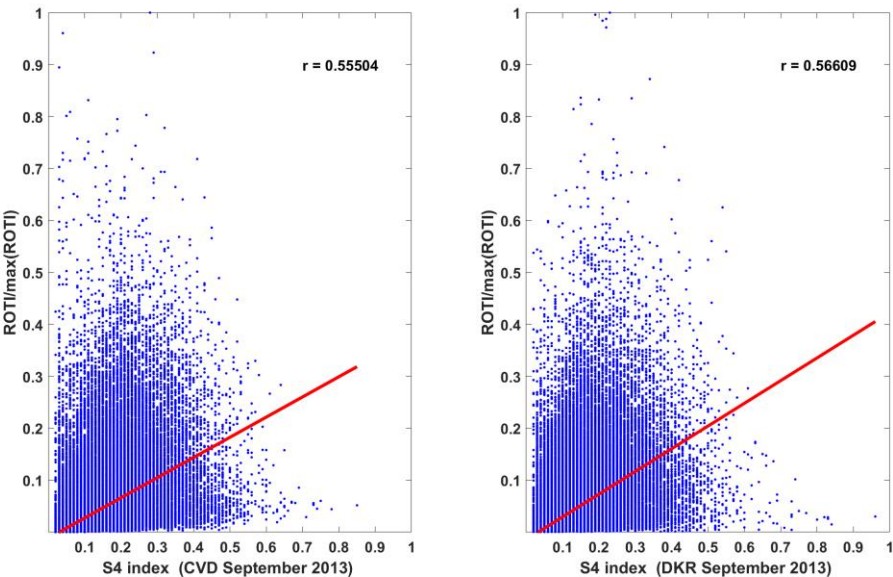

**Figure 4: Correlation between observed ROTI and S4 index for two SCINDA stations in September 2013(left: CVD; right: DKR)**





### 4.2 the dependence of effective velocity

According to equation (9), S4 depends not only on ROTI but also on the effective velocity, leading to the assumption that the
poor correlation between ROTI and S4 is due to the impact of the effective velocity. To validate this assumption, correlation
between observed ROTI and calculated S4 from equation (9) was implemented. To distinguish how the effective velocity
influences the results, the variability of effective velocity in equation (8) is assumed to follow Gaussian distribution, with
parameters $[\mu, \sigma]$. The range of $\mu$ is estimated from equation (8) with available data, and set to be 50 m/s, indicating the
mean value of effective velocity. The $\sigma$ varies according to the ratio of $\mu/\sigma$, indicates the standard deviation   In the
numerical experiments, $\mu/\sigma$ has a range from 0.1 to 3, the larger $\mu/\sigma$ is, the smaller variability of the effective velocity is
assumed, Figure 5 shows the correlation with different $\mu/\sigma$ values; it can be seen that the correlation coefficient decreases
with decreasing $\mu/\sigma$. The correlation coefficient is above 0.9 when $\mu/\sigma$ larger than 0.8; the correlation coefficient dropped
drastically when $\mu/\sigma$ decreased from 0.6 to 0.1. The values are 0.78 when $\mu/\sigma = 0.4$, 0.63 when $\mu/\sigma = 0.2$, and 0.49 when
$\mu/\sigma = 0.1$. It appears that the correlation coefficient depends greatly on the randomness of the effective velocity. Figure 6
shows clearly the influences of $\mu/\sigma$ for both CVD and DKR stations, with the same  $\mu = 50$  set in Figure 5. The correlation
coefficient gradually decreased proportional to the $\mu/\sigma$ for both two stations. The geometric parameters including the
satellite orbit, elevation and azimuth were derived in each calculation from the observables.

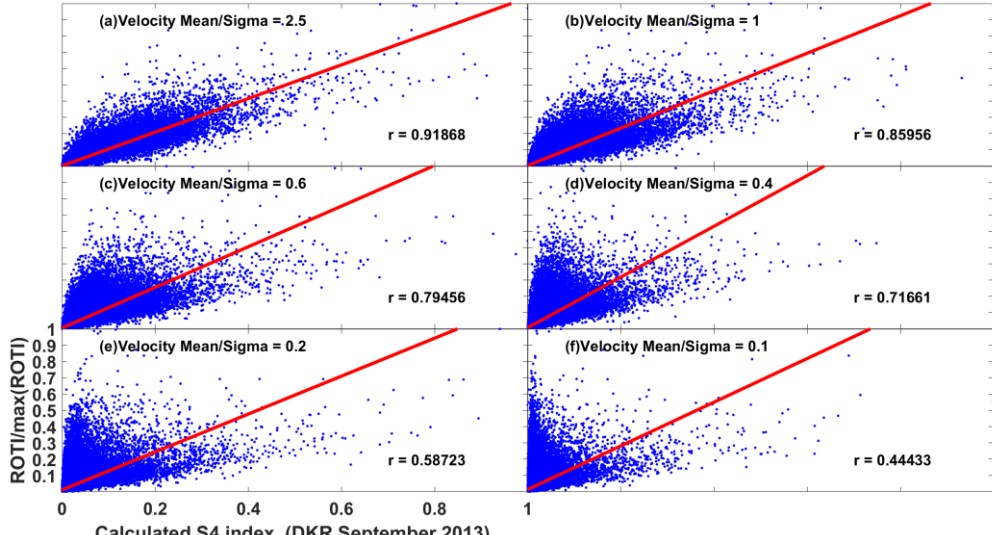

**Figure 5: Correlation between observed ROTI and calculated S4 index.** Correlation between observed ROTI and
calculated S4 index by Equation (9) for one SCINDA stations in September 2013, the different subplots show difference of
effective velocity $\mu/\sigma$, the normalized ROTI was calculated by ROTI/maximum ROTI, and r is the correlation coefficient.
$\mu = 50$ m/s





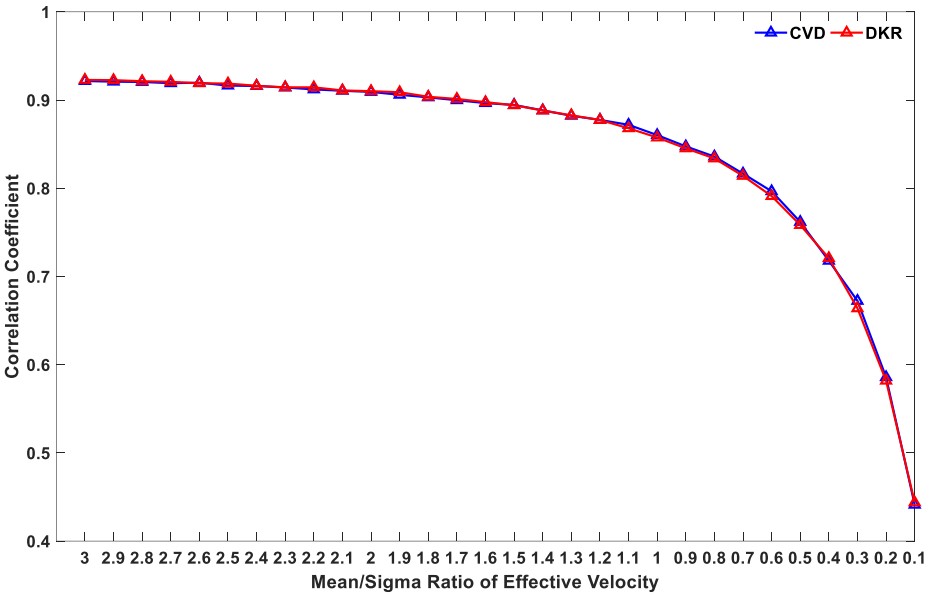

**Figure 6: Correlation between observed ROTI and calculated S4 index for two SCINDA stations**. Correlation between
observed ROTI and calculated S4 index by Equation (9) for two SCINDA stations in September 2013, $\mu/\sigma$ decreases from 3
to 0.1, with interval 0.1; $\mu = 50$ m/s

To further determine the impact of $\mu$ on the correlation coefficient, a range of $\mu$ from 20 m/s to 100 m/s was selected and
tested. Figure 7 shows the correlation coefficient in relation with the effective velocity; the different color curves indicates
different effective velocity, here five values of effective velocities were selected. The $\mu/\sigma$ is set the same range as Figure 6,
from 0.1 to 3, with 0.1 interval. The figure shows that the selection of $\mu$ doesn't influence the correlation coefficient, but
only the $\mu/\sigma$ determines the results.

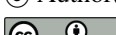


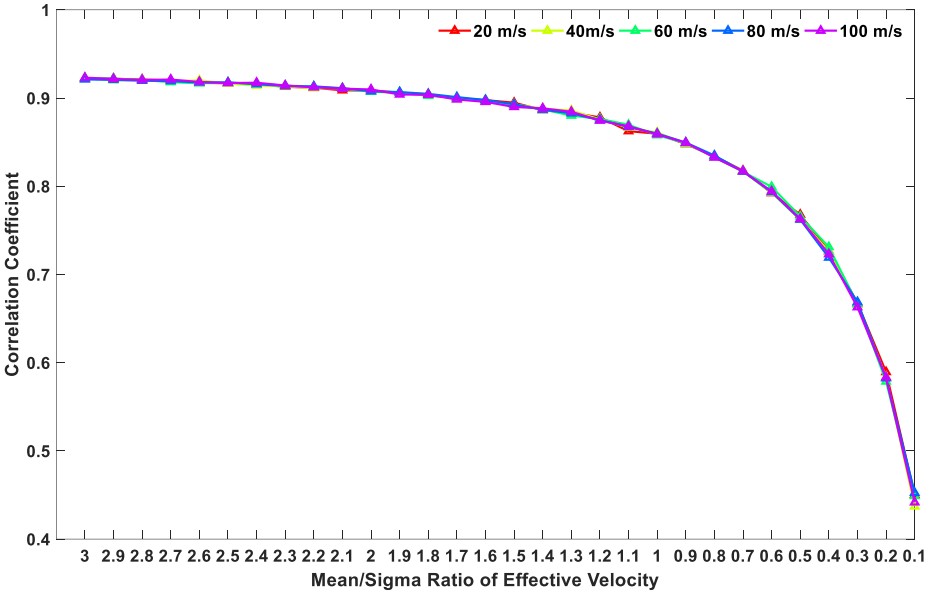

**Figure 7: Correlation between observed ROTI and calculated S4 index by Equation (9) for one SCINDA stations in September 2013, $\mu/\sigma$ decreases from 3 to 0.1, with interval 0.1; μ was selected from 20 m/s to 100 m/s, spanned by 20.**
**The different colors indicate different μ.**

**4.3 the dependence of elevation**

The possible effect of the elevation angle on the correlation between ROTI and S4 was investigated in a similar way. The elevation angles of September 2013 were used with an elevation mask of 30°. A clear exponential distribution was observed. So, in the experiments, the randomness of elevation angle is modeled as exponential distribution, with a mean value $\theta_{mean}$.
The impacts of elevation angle randomness were verified with fixed effective velocity as 50 m/s, also effective velocity was assumed to be independent of the elevation angle. Figure 8 shows the simulation results, with ROTI taken from observables of DKR station. It shows that with increment of $\theta_{mean}$, the correlation coefficient also increases slowly. All the correlation coefficients were above 0.999, showing that the randomness of elevation angle on the distance s plays little role on the results of correlation coefficient. It can be said that under the above assumption, elevation angle doesn't affect the correlation
between calculated S4 from Equation (9) and ROTI.

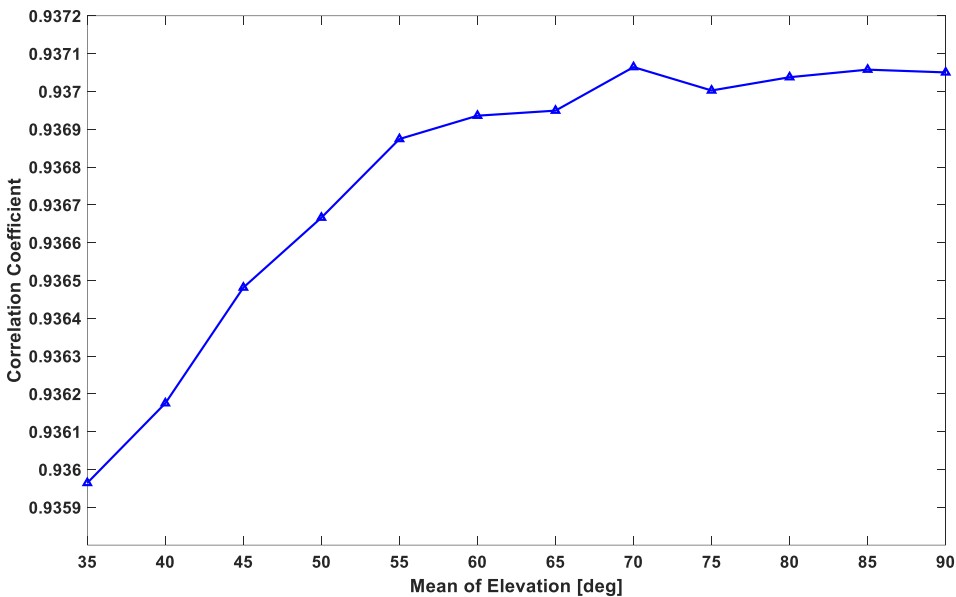

**Figure 8: Correlation between observed ROTI and calculated S4 index by Equation (9) for one SCINDA stations in September 2013; μ = 50 m/s; the mean elevation angle $\theta_{mean}$ varies from 30 to 90, with 5 as interval.**

**5 Discussion**

The single-phase screening model proposed by Rino (1978) indicates that scintillation intensity was strongly related with the scintillation spectrum and Fresnel scale. A series of successive studies further prove that the geometry factors are also crucial to decide the scintillation features (Carrano et al, 2016). Several indices have been used to study scintillation intensity, like S4, $\sigma_\phi$. Even ROTI is often used to investigate scintillation (Veettil and Aquino, 2017; Cherniak and Zakharenkova, 2018). In this study, from equation (9), it is shown that S4 has better correlation with DROTI, rather than ROTI, giving a possible

explanation of the different correlation coefficients between ROTI and S4 found in previous studies. From equation (9) it can be seen that S4 is influenced not only by the irregularity itself represented by the TEC rate of change, but also by the effective velocity, the distance between satellite and receiver and also the propagation path. Carrano et al (2019) pointed out that correlation between ROTI and S4 is affected by the zonal velocity and zenith angle, and that only when zenith angle is less than 20 °can ROTI and S4 achieve a higher correlation. In this work, the randomity and variability of effective velocity

and elevation angle are considered. Figure 5 shows that correlation between ROTI and S4 indeed varies according to the variability of effective velocity. Figure 6 further shows the correlation degradation with decrement of $\mu/\sigma$ , which indicates the variability of the velocity under the Gaussian distribution assumption of effective velocity. Figure 7 confirms that it is the variability of effective velocity that influences the correlation results between ROTI and S4. Results in Figure 6 and Figure 7



suggest that a high correlation between ROTI and S4 can be achieved when $\mu/\sigma > 1$. When poor correlations between
ROTI and S4 were found, a large variability of the effective velocity was the cause. The randomity and variability of
elevation angle shows very weak influence on the correlation coefficient as Figure 8 demonstrates. This doesn't mean that
geometry has very minor impact on the correlation between ROTI and S4, since the effective velocity itself is dependent on
geometry and can be strongly influenced by geometry factors.

The single screen model proposed from equation (5) to equation (7) is in essence the same than the model described by
Carrano et al (2019), but they extended the model to a more complex way by introducing several parameters to describe the
geometry influences on the effective velocity. Carrano et al (2019) also related the scintillation indices with scintillation
spectrum in the signal process aspect. The difference between equation (9) and Du et al (2000) 's model is clear. The
assumption of constant effective velocity in Du's model can easily introduce errors when the variability of effective velocity
is large enough. Thus, Du's model can only work in some specific conditions. Equation (9) also provides clear evidence for
the different physical mechanism represented by ROTI and S4. ROTI indicates the physical irregularities of ionosphere,
while S4 it is related to the radio propagation conditions, showing how the signal will suffer from ionospheric irregular
behavior (indicated by ROTI), but also from the effective velocity. The effective velocity is considered as combination of
zonal drift velocity and the velocity of ionospheric pierce point (Arruda et al, 2006; Chapagain et al, 2013). Thus, in this
study, a clear link between scintillation intensity and the effective velocity behavior has been established by equation (9).

**6 Conclusion**

In this work, the correlation between ROTI and S4 is investigated based on a derived mathematical model represented by
equation (9) and under the assumption of single screening model for trans-ionospheric signal propagation. After taking the
randomity and variability of effective velocity into account, this investigation reveals something new about the correlation
between ROTI and S4 as follows.

(1) The correlation between ROTI and S4 is strongly affected by the variability of the effective velocity, and the
correlation coefficient is degraded when the variability increases. If the randomity of effective velocity is modeled
as Gaussian distribution, a higher correlation coefficient is found when $\mu/\sigma > 1$. At the same time, the value of
effective velocity itself is independent with the correlation coefficient.

(2) The scintillation index S4 depends not onlyon the ionospheric irregularities, but also on the effective velocity and
the trans-ionospheric radio propagation path. In fact, the effective velocity is also impacted by the geometry
factors and ionospheric status itself. S4 is closer linked with DROTI and can be derived when both DROTI and
effective velocity are known.

(3) The physical meaning represented by S4 and ROTI are different. ROTI indicates the internal instability of
ionosphere, and usually is considered as a proxy of ionospheric irregularity. S4 depends on the trans-ionospheric
radio propagation conditions, and is defined by both of ionospheric irregularities and the effective velocity.



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

**Appendix A: Derivation of Equation (9)**

In the appendix, the derivation of Equation (7) in section 2.2 is described as below.

From (7) we get

$$\frac{\mathrm{d}^2 TEC(x)}{dt^2} = \frac{2\pi}{\lambda r_e} \frac{v^2}{\lambda s} \left[ 1 - \frac{I}{I_0} \right],$$
(A-1)

The variation of electron density can be described by the change of total electron content (TEC), then $TEC(x)$ shows the
condition of electron density. $\lambda$ denotes the wavelength of propagating signal, $r_e = 2.8 \times 10^{-15}$ m is the radius of the electron. $I$ is the scintillated signal amplitude, $I_0$ is its standard level under non-scintillation condition. $v$ is a comprehensive velocity in the single phase screen model $z = \frac{(R_e + h_{IPP})\cos(\delta + \theta)}{\cos\theta}$ indicates the slant distance between phase screen ionosphere pierce point and GPS receiver.

Substitute to (A-1), it follows

$\frac{d^2 TEC}{dt^2} = \frac{dROT}{dt} = DROT$,
(A-2)

Substitute (A-2) to the calculation formula of ROTI, it gives as

$$DROTI = \sqrt{\langle DROT^2 \rangle - \langle DROT \rangle^2}$$

The right side of (A-1) can be derived as

$$\sqrt{\left\langle \left( \frac{2\pi}{\lambda r_e} \frac{v^2}{\lambda s} \left[ 1 - \frac{I}{I_0} \right] \right)^2 \right\rangle - \left\langle \frac{2\pi}{\lambda r_e} \frac{v^2}{\lambda s} \left[ 1 - \frac{I}{I_0} \right] \right\rangle^2} = \frac{2\pi v^2}{\lambda^2 r_e s} \sqrt{\left\langle \left[ 1 - \frac{I}{I_0} \right]^2 \right\rangle - \left\langle 1 - \frac{I}{I_0} \right\rangle^2} = \frac{2\pi v^2}{\lambda^2 r_e s} \sqrt{\left\langle 1 - \frac{2I}{I_0} + \frac{I^2}{I_0^2} \right\rangle - \left( 1 - \left\langle \frac{I}{I_0} \right\rangle \right)^2} =$$

$\frac{2\pi v^2}{\lambda^2 r_e s} \sqrt{\left( 1 - \left\langle \frac{2I}{I_0} \right\rangle + \left\langle \frac{I^2}{I_0^2} \right\rangle \right) - \left( 1 - \left\langle \frac{2I}{I_0} \right\rangle + \left\langle \frac{I}{I_0} \right\rangle^2 \right)} = \frac{2\pi v^2}{\lambda^2 r_e s} \sqrt{\left\langle \frac{I^2}{I_0^2} \right\rangle - \left\langle \frac{I}{I_0} \right\rangle^2} = \frac{2\pi v^2}{\lambda^2 r_e s} S_4$ ,
(A-3)

So it gives that





$$DROTI = \frac{2\pi v^2}{\lambda^2 r_e s} S_4 \ , \tag{A-4}$$

Therefore:

$$v^2 = DROTI \cdot \frac{\lambda^2 r_e s}{2\pi S_4}, \tag{A-5}$$

the velocity is related to DROTI variation, amplitude scintillation index S4 and scintillation intensity.

To simplify the equation, we have

$$S_4 = DROTI \cdot \frac{\lambda^2 r_e s}{2\pi v^2}, \tag{A-6}$$