# Peer review of "On the correlation between ROTI and S4"

_Annales Geophysicae, 2019_

## Referee Comment (RC1) · Anonymous Referee #1 · 30 Dec 2019

The work "On the correlation between ROTI and S4" is aimed at the relation between amplitude scintillation intensity measured with S4 and ionospheric variability represented by ROTI. These indices basically refer to different effects (i.e. signal scintillation and TEC fluctuation), but both are widely used for detection of irregularities and thus, such additional, comparative analysis could be helpful for future ionospheric studies. According to the above the selected issue seems to be interesting and worth to be explored. However, after reading the paper, I am not convinced to many aspects of the proposed methodology and scientific soundness, which have to be improved and clarified.

General comments:

1. English should be improved.

[Figure]

2. The used dataset cover time series of S4 (provided, if I am not wrong, directly by Novatel receivers) and ROTI (DROTI). How exactly the latter parameter was computed? Based on TEC from Novatel receivers or from RINEX data and geometry-free combination? Regardless of the method, it should be clearly presented.

3. The equations 8 and 9 connect S4, DROTI and the effective velocity. The two former are usually computed in 1- and 5-minute interval respectively and probably the same time spans were adopted in this work. In my opinion it may be problematic for effective velocity computation (eq.8) during disturbed ionosphere. The issue related to different intervals and its possible impact on velocity determination needs to be clarified.

4. Basically, I do not understand the significance of figures 2 and 3. The first of them depicts time series of S4 not linked with anything. In my opinion it should show series of all parameters used by the authors i.e. S4, ROTI, DROTI, velocity as well as the elevation of satellite. Such example would be good for reader and would provide the first look at the relations between them. Figure 3 demonstrates only that the daily values of effective velocity are not precise. Based on such results it is not clear if this is related to subdaily variations of the estimated parameter or other factor(s). In my opinion the deeper evaluation of this effect is required.

5. I am also not really convinced to analysis given in point 4.2. Instead of use both datasets (S4 and ROTI), the authors compute S4 values assuming the known distribution of effective velocity. In my opinion the obtained correlation depends only on sigma of effective velocity distribution (or mean/sigma as in the presented analysis), what confirms figures 5 and 6. Why the authors did not analyze the agreement between real and computed values of S4? Furthermore, why the mean value of velocity was set to 50 m/s instead of 10 m/s according to the figure 3? (It would not change the correlation, but it is not clear why such value was adopted)

6. The results in figure 7 depict exactly what we can expect. They are simple consequence of equation 9 and are the same for particular mean/sigma ratios.

7. The point 4.3 is again performed with calculated S4 values instead of real. I do not see the significance of such tests, what basically confirms the change of correlation coefficient from 0.936 for low-elevated data to 0.937 for vertical.

Summarizing the above, all tests aimed at correlation between S4 and ROTI should involve real dataset. The discussion and summary have to be corrected accordingly.

Specific comments:

1. In the abstract I would add the information that the analysis concerns low latitudes

2. Line 20, "the more severe scintillation is, the larger scintillation indices will be". Is this sentence necessary?

3. Line 22, "the rate of change of total electron content (TEC) indicated by ROT", ROT is rate of TEC

4. Line 23, "ROT index ROTI" probably should be: ROT index (ROTI)

5. Line 27, "can be achieved" this can be removed

6. Lines 29 and 34, What do you mean by "certain geometry" and "some geographical locations"

7. Line 45, "TEC is derived by extracting ground GNSS observables from SCINDA files" What was the algorithm, please provide more details.

8. Line 49, "Considering that the slant TEC is given by" This describes ROT equation not STEC, please correct this.

9. Lines 77, Please define "the effective velocity". What do you mean by "The velocity is related to ROTI variation, amplitude scintillation index S4 and scintillation intensity"

10. Line 92, "The sampling rate of TEC and scintillation indices is 1 minute, to further calculate ROTI index proposed by Pi et al. (1997), a five-minutes window span was used with 30° elevation mask to exclude multipath interference." According to this,

you have S4 with 1-minute interval based on 50 Hz data and 5-minute ROTI(DROTI) computed using 1 minute data. Please provide the details how these nonhomogeneous data were processed to derive the velocities according to the equation 8.

11. Line 98, "It shows that the effective velocity has daily variabilities." It is not clear based on figure 3 but if so, why it was not showed with subdaily values.

12. Lines 109-112, "in some cases large S4 corresponded to small normalized ROTI", "similarly large S4 with very small ROTI can be found" Please, do not repeat the same information.

13. Line 124, "The $\sigma$ varies according to the ratio of $\mu/\sigma$, indicates the standard deviation" Please, correct this.

14. Line 152, "The elevation angles of September 2013 were used with an elevation mask of 30" Again, please rephrase it.

15. Line 158, "All the correlation coefficients were above 0.999" Such correlation between real values S4 and ROTI cannot be true and is only related to the applied methodology, which is in my opinion wrong. Furthermore, this correlation according the figure 8 equal to 0.937

---

## Referee Comment (RC2) · Anonymous Referee #2 · 18 Feb 2020

The paper is a good contribution to the on going efforts to characterize irregularities and irregularity indices. While the data used is very minimal and one may think that the correlation shown are special cases, its important to note that the idea of effective velocity could be a new variable that can be tested for modeling scintillation measurements. I therefore think the work is significant.

Please also note the supplement to this comment:
https://www.ann-geophys-discuss.net/angeo-2019-147/angeo-2019-147-RC2-supplement.pdf
* * *
[Figure]

**Supplement:**

On the correlation between ROTI and S4

In this work the authors have carried out a study on the correlation between ROTI and S4. The authors have used the variable effective velocity and explored its role on the correlation between the amplitude scintillation index and the TEC rate of change indices. The authors have shown that under low variability conditions, a high correlation coefficient is obtained using effective velocity.

This work is a step forward in attempts to characterize irregularities and scintillation measurements based on the current used indices. The findings based on effective velocity adds a new variable that could be useful in modeling and forecast attempts on scintillation phenomenon which still remain rudimentary. I recommend the paper can be accepted but after minor corrections.

Specific Comments:

1. In line 49, delete slant TEC and change it to ROT.
2. Line 70: Then (5) was rewritten as; delete was and substitute with can.
3. The symbols like Re, hipp are mentioned between line 76-79. The equation they are referring to is presented after line 263 in the appendix. Move lines 76-79 starting from the sentence Re denotes…… to lines 263.
4. In line 123: the parameters $\mu, \delta$ are introduced but not defined anywhere in the document. The authors have assumed that they are well known to readers. Define them for clarity.

Generally the grammar must be revised in the entire document.

Technical errors

The authors should check and ensure that the references are all cited in the document.

For example the following references are in the document but not written in the reference list

-Kintner et al., 2007

-Du et al., 2000

-Bhattacharyya et al 2007

-Rino 1978

Aruda et al., 2006

The following references are not cited in the document

-Aruda et al., 2016 in line 215

-Bhattacharyya et al., 2000 line 220

-Du et al 2001 line 234

-Kintner et al., 2006 line 236

-Li et al., 2007 line 241

-Olwendo et al., 2018 line 243.

-